# Humanization of wildlife gut microbiota in urban environments

**Brian A Dillard[1]\*, Albert K Chung[2], Alex R Gunderson[3], Shane C Campbell-Staton[2], Andrew H Moeller[1]\***

[1]Department of Ecology and Evolutionary Biology, Cornell University, Ithaca, United States; [2]Princeton University, Princeton, NJ, United States; [3]Tulane University, Tulane, United States

**Abstract** Urbanization is rapidly altering Earth's environments, demanding investigation of the impacts on resident wildlife. Here, we show that urban populations of coyotes (*Canis latrans*), crested anole lizards (*Anolis cristatellus*), and white-crowned sparrows (*Zonotrichia leucophrys*) acquire gut microbiota constituents found in humans, including gut bacterial lineages associated with urbanization in humans. Comparisons of urban and rural wildlife and human populations revealed significant convergence of gut microbiota among urban populations relative to rural populations. All bacterial lineages overrepresented in urban wildlife relative to rural wildlife and differentially abundant between urban and rural humans were also overrepresented in urban humans relative to rural humans. Remarkably, the bacterial lineage most overrepresented in urban anoles was a *Bacteroides* sequence variant that was also the most significantly overrepresented in urban human populations. These results indicate parallel effects of urbanization on human and wildlife gut microbiota and suggest spillover of bacteria from humans into wildlife in cities.

## Editor's evaluation

Urbanization has broad impacts on macroecology but its consequences for wildlife microbial ecology remain unclear. Now, Dillard and colleagues provide new data suggesting that humans living in an urban setting may transfer their microbes to wildlife with potentially adverse effects.

**\*For correspondence:**
bd429@cornell.edu (BAD);
ahm226@cornell.edu (AHM)

**Competing interest:** The authors declare that no competing interests exist.

## Introduction

The gut microbial communities of vertebrates tend to reflect their host's phylogenetic histories. Across a diversity of vertebrate clades, the community composition of the gut microbiota is on average more similar within host species than between host species, and microbiota dissimilarity between host species is positively associated with host evolutionary divergence time (*Song et al., 2020*; *Moeller et al., 2017*; *Brooks et al., 2016*; *Muegge et al., 2011*; *Ochman et al., 2010*; *Ley et al., 2008*). However, anthropogenic influences can erode the host-lineage specificity of gut microbiota through a process of humanization, in which hosts acquire microbiota constituents found in humans (*Clayton et al., 2016*; *Houtz et al., 2021*; *Trevelline and Moeller, 2022*). For example, although the specific effects of captivity on microbiota differ among mammalian species (e.g., *Houtz et al., 2021*, *Alberdi et al., 2021*; reviewed in *Diaz and Reese, 2021*), several studies have shown that captive mammals harbor gut microbiota constituents abundant in humans but missing from wild-living conspecific populations (*Clayton et al., 2016*; *Houtz et al., 2021*; *Trevelline and Moeller, 2022*), suggesting transmission from humans. If host species and microbiota have adapted to one another, disruption and replacement of native microbiota may have adverse consequences for host phenotypes and fitness. For example, recent studies in germ-free mice have found that mice seeded with non-native

**eLife digest** Vertebrate species, such as reptiles, birds or mammals, harbour distinct communities of microbes in their digestive systems. These miniature ecosystems – also known as microbiomes – are unique to each owner and species, reflecting their diverse lifestyles and evolutionary history.

Urbanisation can disrupt these delicate intestinal communities. Humans and other animals living in cities have different gut microbes to their counterparts living in rural areas. And captive species in homes and zoos often acquire human gut bacteria in their digestive systems, which can lead to health problems in these animals. So far, it has been unclear whether such a humanization of gut bacteria also affects wild animals living in and around cities.

To investigate this further, Dillard et al. compared the gut microbes of wild reptiles, birds, and mammals living in close contact with humans in North America, such as coyotes, crested anole lizards and white-crowned sparrows. DNA sequencing showed that in urban environments, the composition of gut bacteria living in all three wildlife species resembled the ones in humans. The types of bacteria overrepresented in the guts of urban humans were also overrepresented in urban wildlife.

This suggests that urbanization can affect the composition of gut bacteria in wildlife species by disrupting or replacing portions of their microbiome. The reason for this pattern is unclear. It is possible that humans might be sharing their gut microbes directly with city animals, or that a human-like diet is causing the change. Given the role that gut microbes play in health and disease, it is important to find out whether these changes cause the animals any harm.

microbiota displayed stunted immunological development and growth rates relative to house mice seeded with native, house-mouse microbiota (*Chung et al., 2012*; *Moeller et al., 2019*) (although certain non-native, e.g., human, gut microbiota have been shown to provide growth and immune benefits to mice relative to other non-native microbiota or the germ-free state; e.g., *Turnbaugh et al., 2006*; *Round and Mazmanian, 2010*). Similarly, disruption of gut microbiota in captive mammals has been implicated in the gastrointestinal disorders often experienced by captive hosts (*Clayton et al., 2016*; *Diaz and Reese, 2021*; *McKenzie et al., 2017*), and efforts to restore wild microbiota (e.g., through fecal microbiota transplantation) to improve the health of captive animals has seen success in some cases (e.g., *Koeppel et al., 2006*; reviewed in *Diaz and Reese, 2021*).

With the influence of humans on ecosystems spreading globally, it is necessary to understand the effects on microbiota of vertebrate wildlife, including the possibility of microbiota convergence with humans. Of particular importance are the effects of urbanization, which are escalating at accelerating rates (*Kalnay and Cai, 2003*; *Sun et al., 2020*). Gut-microbiota composition differs between urban and rural settings in diverse species of vertebrate wildlife (*Teyssier et al., 2020*; *Sugden et al., 2020*; *Berlow et al., 2021*). However, the extent to which the gut microbiota of urban wildlife converge with those of humans has not yet been explored.

## Results and discussion

To test for humanization of wildlife gut microbiota in urban environments, we compared the gut microbiota of three vertebrate species—coyotes (*Canis latrans*), anoles (*Anolis cristatellus* and *Anolis* spp.), and white-crowned sparrows (*Zonotrichia leucophrys*)—and humans in urban and rural settings. Anoles, coyotes, and sparrows maintain both urban and rural populations throughout North America and represent model systems for study of impacts of urbanization on vertebrate biology (*Berlow et al., 2021*; *Campbell-Staton et al., 2020*; *Tigas et al., 2002*). We analyzed 492 fecal microbiota profiles from 94 crested anoles, 33 anole lizards of other species (*Anolis* spp.) (i.e., non-*cristatellus* anoles), 78 coyotes, 87 sparrows, and 487 adult humans. We sampled anoles in the city of Mayagüez on the western coast of Puerto Rico and across an eastern longitudinal transect through less densely populated areas in Quemado into the rural areas in Maricao (*Figure 1—figure supplement 1*). Coyote data derived from *Sugden et al., 2020* were sampled in the city of Edmonton, Alberta, Canada, and rural areas around Leduc. Sparrow data derived from *Berlow et al., 2021* included urban and rural sampling sites around San Francisco, California, United States. Human microbiota data were combined from three studies, including urban populations in the United States and rural populations

in Malawi and Venezuela (*Yatsunenko et al., 2012*) as well as urban, semiurban (i.e., suburban), and rural populations in Cameroon (*Lokmer et al., 2020*) and Tibet (*Li et al., 2018*). Sample metadata are presented in *Supplementary file 1*. A total of 49,281 Amplicon Sequence Variants (ASVs) were observed, and their relative abundances are presented in *Supplementary file 2* and taxonomic assignments against the Silva 138 database in *Supplementary file 3*. Taxonomic barplots for all sample groups are presented in *Figure 1—figure supplement 2*.

Next, we calculated all microbiota dissimilarities (binary Sorensen–Dice and Bray–Curtis) between pairs of samples (*Bolyen et al., 2019*). Sorensen–Dice was used to test whether the community memberships of gut microbiota of urban wildlife and humans have converged relative to those of rural wildlife and humans. PERMANOVA and PERMDISP indicated significant differences in gut-microbiota composition among wildlife populations in different sampling locations ($p < 0.01$ for each PERMANOVA comparison and $p > 0.05$ for each PERMDISP comparison), and random forest models were able to discriminate rural and urban ASV profiles for each wildlife species (*Figure 1—figure supplement 3*). Moreover, adonis2 PERMANOVA indicated independent effects of host species identity ($R^2 = 0.274$, p value = 0.001), host population ($R^2 = 0.0743$, p value = 0.001), and whether the population was urban or rural ($R^2 = 0.0765$, p value = 0.001). Similar effects were observed in analyses based on weighted dissimilarities (species identity $R^2 = 0.192$, p value = 0.001; host population $R^2 = 0.0778$, p value = 0.001; rural/urban $R^2 = 0.0769$, p value = 0.001).

Using pairwise beta diversity similarities, we then tested for each wildlife species whether the mean similarity between urban wildlife and human microbiota was significantly higher than that between rural wildlife and human microbiota. Results indicated that microbiota similarity to humans was significantly higher in urban settings than in rural settings across all wildlife species examined (*Figure 1A-C*) (nonparametric pv alue <0.05 in 51/54 comparisons between pairs of groups; *Supplementary file 4*). Moreover, anole microbiota profiles displayed a gradient of similarity to humans that recapitulated the longitudinal transect from Mayagüez (urban) to Maricao (rural) (*Figure 1B*; *Figure 1—figure supplement 1*). The increased similarity of urban wildlife and human microbiota was evident in principal coordinates plots of urban and rural conspecific wildlife populations and urban human populations from the United States (i.e., the urban human populations closest geographically to the wildlife populations) (*Figure 1D and E*) as well as principal coordinates plots of all populations (*Figure 1—figure supplement 4*). Note that most urban wildlife clustered more closely with rural conspecifics than they did with humans, reflecting predominant effects of host-species identity in both urban and rural environments. We also observed that microbiota of urban humans and urban wildlife were more similar on average than were microbiota of rural humans and urban wildlife (*Figure 1A-C*, *Supplementary file 4*) and that microbiota of individual hosts of the same species were significantly more similar on average within urban populations than within rural populations (*Figure 1—figure supplement 5*). Moreover, the microbiota of urban populations of different wildlife species were more similar on average than urban and rural populations in 2/3 comparisons between pairs of wildlife species (*Figure 1—figure supplement 6*). Similar results were observed for analyses based on Bray–Curtis similarities (*Supplementary file 5*). Cumulatively, these analyses indicated that the gut microbiota of distantly related vertebrates (anoles, coyotes, and sparrows) have converged compositionally with human gut microbiota in urban environments relative to the gut microbiota of rural conspecifics.

The convergence of gut-microbiota community memberships between wildlife and humans in urban settings could result from parallel loss or gain of ASVs. Alpha diversity (Chao1 and Shannon entropy) tended to be lower on average in urban human populations than in rural human populations (nonparametric p value <0.05 in 2/3 sets of comparisons between urban and rural human populations; *Figure 1—figure supplement 7*), consistent with loss of ASVs. However, the opposite trend was observed in wildlife (nonparametric p value <0.05 in 2/6 sets of comparisons between urban and rural wildlife populations), consistent with gain of ASVs. To enable identification of the specific ASVs underlying the convergence among urban human and urban wildlife gut microbiota, we identified all ASVs shared by urban wildlife populations and humans to the exclusion of rural wildlife populations. These analyses revealed that more ASVs were shared by urban wildlife and humans to the exclusion of rural wildlife than by rural wildlife and humans to the exclusion of urban wildlife (*Supplementary file 6*), indicative of convergence of urban wildlife microbiota with human microbiota relative to rural wildlife microbiota.

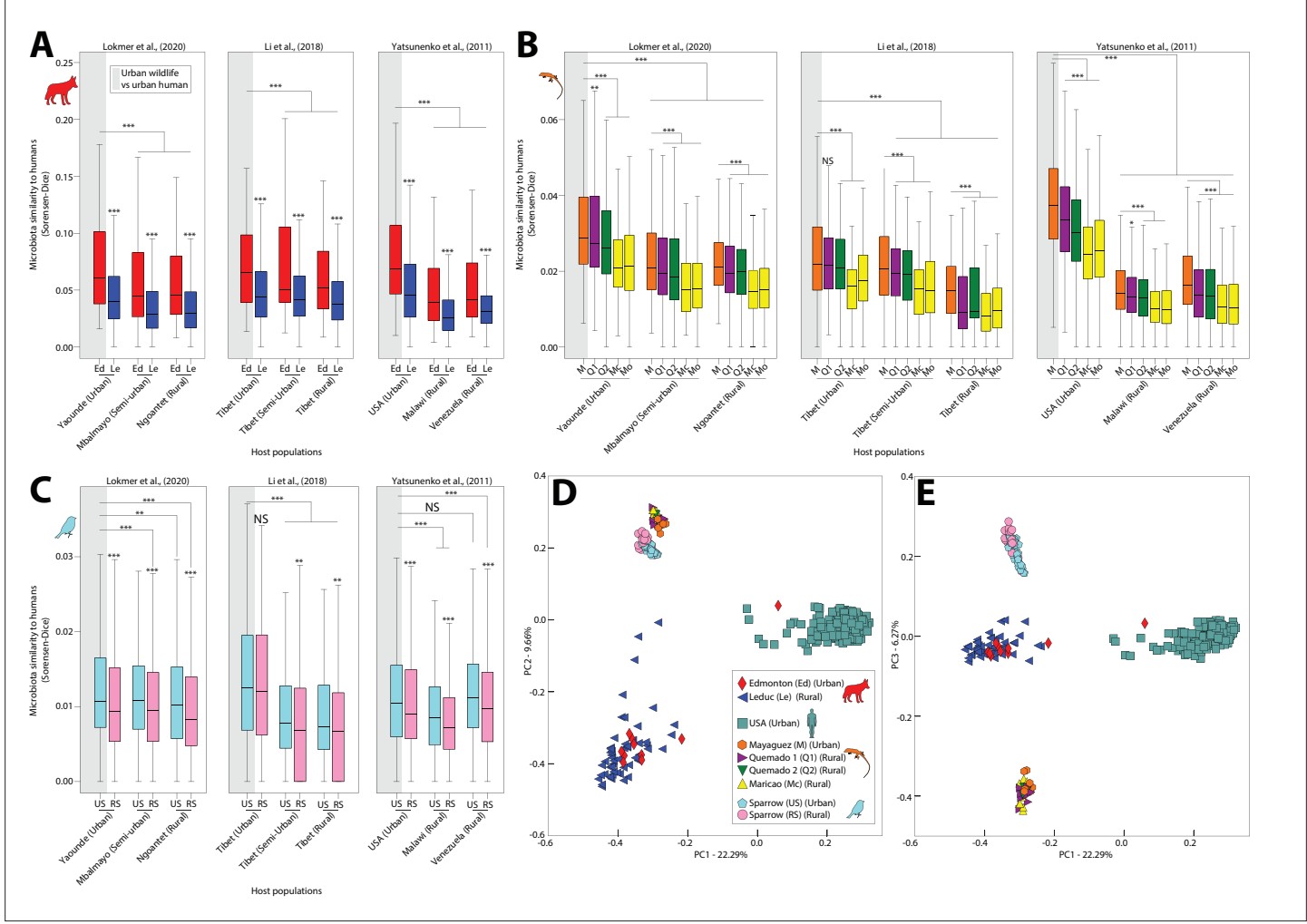

**Figure 1.** Humanization of urban anole and coyote gut microbiota. Boxplots in (**A–C**) show microbiota similarities (binary Sorensen–Dice) between wildlife and human populations. Each panel (A–C) contains results derived from comparisons including a single wildlife species, and each plot within each panel contains results derived from comparisons including a single human datasets (*Lokmer et al., 2020*; *Li et al., 2018*; or *Yatsunenko et al., 2012*). Each box corresponds to comparisons including a single wildlife population as indicated by the key in (**D**). Gray shading behind single boxplot in each plot indicates the comparison between urban wildlife and urban human populations. Boxplots display median and interquartile range. Asterisks directly above boxplots indicate significant differences of dissimilarity to a single human population between urban and rural conspecific wildlife populations. Other asterisks indicate significant differences to the comparison between urban wildlife and urban humans (leftmost boxplot in each plot). p values were calculated from nonparametric Monte Carlo permutation tests; NS p > 0.05; *p < 0.05; **p < 0.01; ***p = 0.001. Principal coordinates analysis plots in (**D**) and (**E**) show patterns of dissimilarities among anole, coyote, and human (USA adults) gut microbiota profiles based on binary Sorensen–Dice dissimilarities. Each point represents the gut-microbiota profile of an individual anole, coyote, sparrow, or human, as indicated by the key in (**D**).

The online version of this article includes the following figure supplement(s) for figure 1:

**Figure supplement 1.** Map of wildlife sampling locations.

**Figure supplement 2.** Taxonomic profiles of gut microbiota in urban and nonurban locations.

**Figure supplement 3.** Random forest analyses discriminate urban and rural wildlife populations.

**Figure supplement 4.** Principal coordinates plots of all samples.

**Figure supplement 5.** Differences among intrapopulation beta diversity between urban and nonurban populations.

**Figure supplement 6.** Tests for convergence of gut microbiota between wildlife species in urban environments.

**Figure supplement 7.** Differences in alpha diversity between urban and nonurban populations.

In addition, to identify the ASVs significantly overrepresented in urban populations, we used ANCOM (*Mandal et al., 2015*) to test for differentially abundant ASVs between urban and rural populations for each wildlife species and human dataset (*Figure 2*). Lists of ASVs differentially abundant between rural and urban samples for each wildlife species and human dataset are presented in *Supplementary file 7*. We identified the ASVs that were differentially abundant between urban and rural populations in at least one wildlife species and at least one human dataset (*Figure 2*). These analyses revealed ASVs that were differentially abundant between urban and rural populations in both coyotes and humans and in both anoles and humans, but none in both sparrows and humans. Clustering of ASVs near zero in *Figure 2* suggests limited power, yet all ASVs that were overrepresented in urban wildlife and detected in humans showed the parallel shifts in relative abundance between urban and rural populations of humans (*Figure 2C–F*) (*Supplementary file 7*). These results indicate parallel effects of urbanization on the relative abundances of certain ASVs in both humans and individual wildlife species, similar to the parallels that have been reported between the effects of urbanization and domestication on the microbiota of humans and wildlife, respectively (*Reese et al., 2021*). These ASVs included a predominant human *Bacteroides* commensal that constituted up to 15% of human gut microbiota in urban environments (*Figure 2C*) and was the single most significantly overrepresented ASV in urban humans (*Figure 2—figure supplement 3*). This ASV was also identified as overrepresented in urban populations in analyses that considered all wildlife populations simultaneously with host species as a covariate (*Supplementary file 7*; *Figure 2—figure supplement 4*).

Overall, this study demonstrates that gut microbiota of urban populations of wildlife species were more similar to human gut microbiota than were the gut microbiota of rural, conspecific wildlife populations. Interestingly, this convergence was evident even though the urban human and wildlife populations examined resided in different geographic locations. One possible explanation for these results is that urban environments imposed parallel selective pressures on wildlife and human microbiota relative to rural environments. For example, parallel dietary shifts in urban environments may select for common sets of ASVs in humans and wildlife. Previous work in humans has shown that *Bacteroides* is positively associated with diets high in animal fat and protein (*Wu et al., 2011*), but the extent to which this or other dietary shifts experienced by humans in urban environments are shared by the urban wildlife sampled here is not known. Our results motivate future profiling (e.g., metabarcoding) of diets and gut-microbiota composition in urban and rural populations.

A nonmutually exclusive explanation for our results is bacterial spillover from humans into wildlife in cities. Testing the rates at which bacterial lineages transmit between humans and wildlife in cities will require further strain-level profiling of the gut microbiota of these hosts sampled in shared urban environments. Increased microbiota similarity in urban environments could not be readily explained by microbiota transmission from wildlife into humans in cities, because none of the wildlife species examined here occur in two of the regions where both urban and rural humans were sampled (i.e., Cameroon and Tibet). Close contact among hosts of the same species can generate social microbiomes—microbial metacommunities formed by microbial transmission along host social networks (*Sarkar et al., 2020*). Previous studies have also indicated that transmission of the gut microbiota can occur between distantly related wild-living mammalian species when they come into close contact, such as predator–prey interactions (*Moeller et al., 2017*). Here, we observed that urban wildlife shared more microbiota constituents with human populations, including rural human populations residing on different continents, than did rural wildlife. These observations imply the existence of gut microbial transmission routes among humans and distantly related vertebrate species in urban environments, motivating the need to investigate consequences of urban-associated microbiota changes for wildlife health and fitness.

## Materials and methods
### Sample collection

Fecal samples were collected from anoles (*A. cristatellus*) in four sampling locations shown in *Figure 1A*. Animals were caught with lassos and placed into clean plastic bags for fecal collection. All sampling was approved by the institutional animal care and use committee at Tulane University. All fecal material was snap frozen upon collection and stored at −80°C. Sample sizes were determined by catch success over a 9-day period from July 14, 2019 to July 23, 2019.

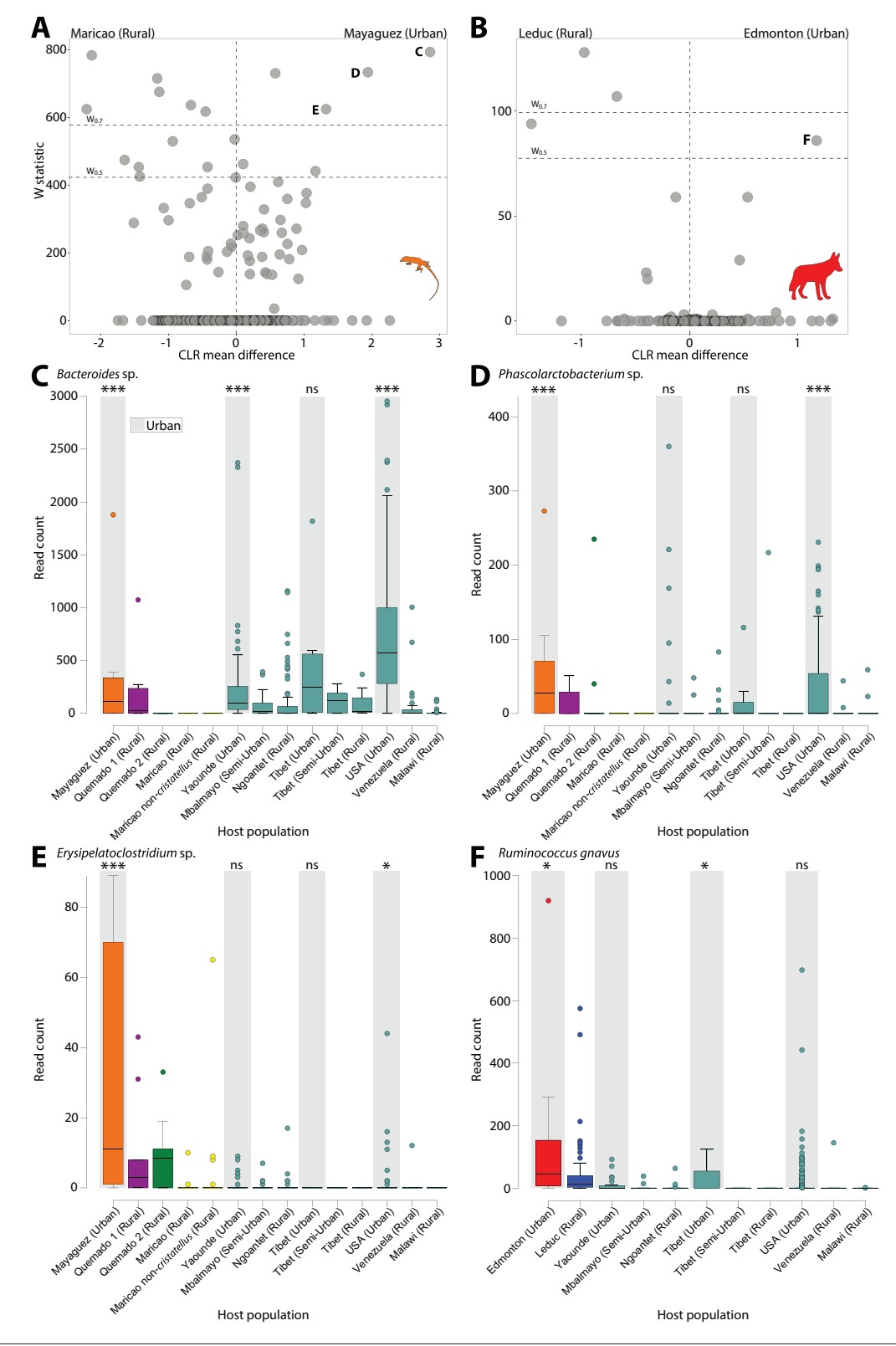

**Figure 2.** Differentially abundant Amplicon Sequence Variants (ASVs) between populations in urban and natural settings. Volcano plots display the results of ANCOM tests for ASVs that were differentially abundant between urban and rural populations of anoles (**A**) and coyotes (**B**) and between urban and rural populations in at least one human dataset. ANCOM results from sparrows showed no significant overlap with human datasets and are

*Figure 2 continued on next page*

*Figure 2 continued*

presented in *Figure 2—figure supplement 1*. Each point represents an ASV, with the *x*-axis denoting the centered log ratio mean difference of the relative abundance of the ASV between urban and rural populations as calculated by ANCOM (*Teyssier et al., 2020*). The *y*-axis indicates the ANCOM test statistic (*W*), and horizontal dashed lines indicate the significance levels *W* = 0.7 and *W* = 0.5. Panel (**A**) displays results of ANCOM tests comparing Mayagüez and Maricao anoles. Results of ANCOM tests comparing Mayagüez anoles with anoles from the other two sampling locations are presented in *Figure 2—figure supplement 2*. Human ANCOM results are shown in *Figure 2—figure supplement 3*. Boxplots in (**C**) – (**E**) show the relative abundances of ASVs that were significantly overrepresented in Mayagüez (urban) anoles relative to Maricao anoles based on ANCOM analyses shown in (**A**) and differentially abundant in at least one human dataset. Boxplots in (**F**) show the relative abundances of the single ASV that was significantly overrepresented in Edmonton (urban) coyotes relative to Leduc coyotes based on ANCOM analyses shown in (**B**) and differentially abundant in at least one human dataset. Boxplots display median and interquartile range. Note that each ASV displays parallel shifts in relative abundance in urban wildlife populations and humans relative to rural populations. Significant differences between ASV relative abundance between urban and rural populations for each dataset are indicated by asterisks above the boxplot for each urban population; ns *W* < 0.5; \**W* > 0.5; \*\*\**W* > 0.7.

The online version of this article includes the following figure supplement(s) for figure 2:

**Figure supplement 1.** Differentially abundant Amplicon Sequence Variants (ASVs) between anole populations in urban and natural settings.

**Figure supplement 2.** Differentially abundant Amplicon Sequence Variants (ASVs) between anole populations in urban and natural settings.

**Figure supplement 3.** Differentially abundant Amplicon Sequence Variants (ASVs) between human populations.

**Figure supplement 4.** Differentially abundant Amplicon Sequence Variants (ASVs) between wildlife populations with host species as a covariate.

## DNA extractions, library preparation, and sequencing

DNA was extracted from each anole fecal pellets with a bead beating procedure based on the Qiagen PowerLyzer kit. The V4–V5 region (515F 926R primer pair) of 16S rRNA gene was amplified from all DNAs in duplicate with the high-fidelity Phusion polymerase and combined as described by *Comeau et al., 2017*. Following library preparation, 16S rRNA gene libraries were pooled in equimolar amounts and sequenced on a single lane of Illumina MiSeq using 300 + 300 bp paired-end V3 chemistry following protocols of *Mandal et al., 2015*.

## Quality filtering, sequence processing, and taxonomic assignments

Raw fastq files generated from anole fecal microbiota libraries were uploaded to the qiita webserver (https://qiita.ucsd.edu/) and combined with publicly available fastq files containing reads corresponding to 16S rRNA gene V4 sequences from the gut microbiota of adult humans, coyotes, and sparrows. Reads detected in negative control wells were removed prior to downstream analyses. Raw reads were filtered for quality using split libraries and trimmed to a common length of 100 bp to enable comparisons across datasets. Amplicon Sequence Variants were called using deblur as implemented in qiita using default parameters. All ASVs were assigned to taxonomic ranks against the Silva 138 database using the taxonomy command in QIIME2. Samples whose read depths were more than three standard deviations below the mean of the samples' group or for which >75% of the reads belonged to a single ASV were removed prior to downstream analyses. For each analysis, samples were rarefied to a common depth of 90% of the minimum read depth of samples included in the analysis in order to enable direct comparisons of alpha and beta diversity among samples.

## Beta and alpha diversity analyses

Beta diversities (binary Sorensen–Dice and Bray–Curtis dissimilarities) were calculated between all pairs of samples and for each individual sample with QIIME2 (*Bolyen et al., 2019*). Nonphylogenetic measures were employed to increase power to detect acquisition by urban wildlife of ASVs found in humans but closely related to ASVs already present in rural wildlife. Alpha diversities (Chao1 and Shannon) were calculated for each sample and plotted by sampling location in QIIME2. Comparisons of beta diversities between sample groups were conducted with anosim using Kruskal–Wallis tests and PEMANOVA/PERMDISP (adonis) in QIIME2. For PERMANOVA and PERMDISP analyses,

sample groups included urban and rural sparrows, Mayaguez anoles, Quemado 1 anoles, Quemado 2 anoles, Maricao anoles, urban coyotes, periurban coyotes, USA humans, Venezuela humans, Malawi humans, and urban, semiurban, and rural humans from Cameroon and Tibet. Tests of differences in mean beta diversity between pairs of comparisons of sample groups were conducted with nonparametric tests based on 999 Monte Carlo permutations of the beta diversity matrix as implemented in QIIME1 (*Caporaso et al., 2010*). For adonis analyses, beta diversity matrices were modeled as a linear combination of host-species identity, host population, and urban status using the formula structure: beta diversity ~ host-species identity + host population + urban status. Significance of comparisons of alpha diversities between sample groups was assessed with Kruskal–Wallis tests in QIIME2. Beta diversities were visualized using PCoA as implemented in QIIME2.

## Statistical analyses of beta diversity

The hypothesis that the gut microbiota of urban wildlife displayed increased compositional similarity to human gut microbiota relative to the gut microbiota of nonurban wildlife was assessed based on comparisons of binary Sorensen–Dice and Bray–Curtis dissimilarities with Monte Carlo permutation tests. Specifically, these tests assessed whether beta diversity between urban wildlife and humans was significantly lower than that between nonurban wildlife and humans. These tests included 999 permutations of the beta diversity matrices as implemented in make_distance_boxplots.py in QIIME1. Random forest models were trained on 50% of the samples included in each comparison using 'qiime sample-classifier classify-samples' in QIIME2 with the following parameters: `--p-test-size 0.5; --p-step 0.05; --p-cv 5; --p-n-estimators 100; --p-estimator RandomForestClassifier; --p-optimize-feature-selection False; --p-parameter-tuning False; --p-palette sirocco`.

## Sensitivity of coyote results to outliers and disparity in sample size

A single coyote sample (SRR8774451) displayed more similar microbiota composition to humans than other samples. To test whether this potential outlier was underlying results, we reperformed all analyses of beta diversity with this sample removed. In addition, to test whether coyote results were driven by uneven sampling between rural and urban populations, we reperformed all analyses of beta diversity with five random subsamples of ten rural coyotes. Only significant results observed in all three sets of analyses (i.e., total dataset, SRR8774451 removed, and subsampled) are reported.

## ASVs shared by urban and rural wildlife and humans

To test whether more ASVs were shared by urban wildlife and humans to the exclusion of rural wildlife than by rural wildlife and humans to the exclusion of urban wildlife, we identified all ASVs displaying these distributions for every wildlife species. In addition, to account for differences in sample size between urban and rural populations, we randomly sampled a subset of rural individuals for each wildlife species equal to the number of urban individuals for that species. We then identified the ASVs shared by urban wildlife and humans to the exclusion of rural wildlife than by rural wildlife and humans to the exclusion of urban wildlife in these random subsamples. We conducted this subsampling analysis 10 times such that the results did not depend on a particular subset of samples. For anole comparisons, the *cristatellus* Maricao population was used as the 'rural' anole population. All humans were included for all comparisons.

## Differential abundance testing and visualization

Analysis of compositions of microbiomes with bias correction (ANCOM) (*Campbell-Staton et al., 2020*) was employed to test for differentially abundant ASVs between sample groups. For these analyses, the ASV table and metadata were imported as a Phyloseq (*McMurdie and Holmes, 2013*) object in R. Sample groups considered were urban versus nonurban anoles, urban versus nonurban coyotes, urban versus rural sparrows, and urban versus rural humans from each of the human datasets. In addition, we also tested for ASVs that were differentially abundant between rural and urban populations across all wildlife species, using host species as a covariate in the model. Centered log ration transformations were conducted using ANCOM as described by *Campbell-Staton et al., 2020* and implemented by scripts available at https://github.com/FrederickHuangLin/ANCOM, copy archived at swh:1:rev:c40aafdc22d65a04645d1b31dfea7c23a0d4d4dc (*Dillard, 2022a*) Differentially abundant

ASVs were visualized using ggplot. The specific scripts used in our analyses are available at https://github.com/briandill2/MicrobiotaUrbanization, copy archived at swh:1:rev:85220569457459d2c-1c9dd4d467236eab75e2898 (*Dillard, 2022b*). Boxplots of ASV relative abundances were made with ggplot. For boxplots of ASV relative abundances, we identified each ASV that was significantly over-represented in a urban wildlife population relative to a conspecific rural wildlife population as well as at least one urban human population relative to the corresponding rural human population.

## Additional information

### Funding

| Funder | Grant reference number | Author |
|---|---|---|
| National Institute of General Medical Sciences | R35 GM138284 | Andrew H Moeller |

The funders had no role in study design, data collection, and interpretation, or the decision to submit the work for publication.

### Author contributions

Brian A Dillard, Formal analysis, Investigation, Methodology, Validation, Visualization, Writing - original draft, Writing - review and editing; Albert K Chung, Data curation, Investigation, Methodology, Resources, Writing - review and editing; Alex R Gunderson, Shane C Campbell-Staton, Conceptualization, Data curation, Investigation, Methodology, Project administration, Resources, Writing - review and editing; Andrew H Moeller, Conceptualization, Data curation, Formal analysis, Funding acquisition, Investigation, Methodology, Project administration, Resources, Supervision, Validation, Visualization, Writing - original draft, Writing - review and editing

### Author ORCIDs

Brian A Dillard (ID) http://orcid.org/0000-0003-1845-2980
Andrew H Moeller (ID) http://orcid.org/0000-0002-8377-4647

### Decision letter and Author response

Decision letter https://doi.org/10.7554/eLife.76381.sa1
Author response https://doi.org/10.7554/eLife.76381.sa2

## Additional files

### Supplementary files

- Supplementary file 1. Metadata for all samples.
- Supplementary file 2. Amplicon Sequence Variant (ASV) relative abundances across all samples.
- Supplementary file 3. Taxonomic assignments of all Amplicon Sequence Variants (ASVs).
- Supplementary file 4. Statistics for pairwise beta diversity comparisons based on Sorensen–Dice.
- Supplementary file 5. Statistics for pairwise beta diversity comparisons based on Bray–Curtis.
- Supplementary file 6. Amplicon Sequence Variants (ASVs) shared by urban wildlife and humans but not by rural conspecific wildlife.
- Supplementary file 7. ANCOM statistics from comparisons of urban and rural populations.
- Transparent reporting form

### Data availability

Sequencing data have been deposited in Data Dryad at https://doi.org/10.5061/dryad.dfn2z353d.

The following dataset was generated:

| Author(s) | Year | Dataset title | Dataset URL | Database and Identifier |
|---|---|---|---|---|
| Moeller AH | 2022 | Humanization of wildlife gut microbiota in urban environments | https://dx.doi.org/10.5061/dryad.dfn2z353d | Dryad Digital Repository, 10.5061/dryad.dfn2z353d |

The following previously published datasets were used:

| Author(s) | Year | Dataset title | Dataset URL | Database and Identifier |
|---|---|---|---|---|
| Yatsunenko T, Rey FE, Manary MJ, Trehan I, Dominguez-Bello MG | 2012 | Human gut microbiome differentiation viewed across cultures, ages and families | https://qiita.ucsd.edu/study/description/850 | qiime, 850 |
| Sugden S, Sanderson D, Ford K, Stein LY | 2020 | An altered microbiome in urban coyotes mediates relationships between anthropogenic diet and poor health | https://www.ncbi.nlm.nih.gov/bioproject/PRJNA528764/ | NCBI BioProject, PRJNA528764 |

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
