## [Editor Report]

Urbanization has broad impacts on macroecology but its consequences for wildlife microbial ecology remain unclear. Now, Dillard and colleagues provide new data suggesting that humans living in an urban setting may transfer their microbes to wildlife with potentially adverse effects.

---

## [Decision Letter]

**Decision letter after peer review:**

Thank you for submitting your article "Humanization of wildlife gut microbiota in urban environments" for consideration by *eLife*. Your article has been reviewed by 3 peer reviewers, including Peter J Turnbaugh as Reviewing Editor and Reviewer #1, and the evaluation has been overseen by George Perry as the Senior Editor. The following individuals involved in review of your submission have agreed to reveal their identity: Aspen T. Reese (Reviewer #2); Kevin Kohl (Reviewer #3).

Essential revisions:

1) Expand the human datasets analyzed. The current data doesn't seem that well suited to look at urbanization vs. other confounding factors. If possible, it would be great to include other wildlife urban/rural comparisons and other human datasets which more explicitly get at urbanization. Potential comparator datasets include:

https://link.springer.com/article/10.1007/s00248-020-01569-8

https://www.sciencedirect.com/science/article/abs/pii/S0048969717323781

https://onlinelibrary.wiley.com/doi/full/10.1111/mec.15240

https://pubmed.ncbi.nlm.nih.gov/30439937/

2) Address the various other points itemized below, including addressing the limitations and caveats of these data.

*Reviewer #1 (Recommendations for the authors):*

As mentioned in the discussion, the addition of metagenomics would greatly strengthen the conclusions that can be made. Alternatively, the authors could use qPCR, FISH, selective culture, or other methods to strengthen the interpretation that the same bacteria are shared between host species.

It is also important to rule out any technical explanations for overlap in ASV content between species, including barcode hopping, cross-contamination, or other issues. More discussion should be included as to how these samples were sequenced and what control samples were included to rule out these artifacts.

There seems to be a conceptual issue in the degree to which one should be surprised to find overlapping bacterial species between host species. As Drs. Moeller, Ley, and others have shown, the phylogenetic signal requires sophisticated analyses and is only true for subsets of the gut microbiota. It would help to assess the range of ASV overlap between humans and other species to see if the observed hit rate in this study would be expected by chance.

I also didn't follow the logic that transfer events would result in differential abundance as a function of urbanization. Isn't it possible that bacteria could be consistently transferred from humans to coyotes in both urban and rural settings? What if a transfer event only happens in urban settings but is rare, resulting in a non-significant p-value? It's important to provide a clearer rationale for why each analysis was done and whether it answers the hypothesis.

*Reviewer #2 (Recommendations for the authors):*

Here I note some particular ways to address shortcomings noted in the Public Review.

– The human data used here were originally collected to analyze industrialization effects rather than urbanization effects and span populations that vary in many dimensions. While obviously related urbanization and industrialization transitions are distinct so it is impossible to say that urbanization is really what underlies the effects here. No good datasets exist comparing urban and rural populations in North America (which would be ideal because that is where all the wildlife data is from) but there are within country analyses for Cameroon (Lokmer et al., Sci Rep 2020), Nigeria (Ayeni et al., Cell Rep 2018), and India (Das et al., Sci Rep 2018). Presumably if the urbanization signal is thought to be generalizable it should still show up in studies from outside North America so why not use those? Less ideal but perhaps relevant are the gradient of industrialization presented in Jha et al., (2018 PLoS Bio) and the comparison of Indigenous Canadians in Montreal and Inuit villages (Girard et al., mSphere 2017) at least control for geography when investigating lifestyle transitions. Perhaps some combination of these datasets would allow for a more robust test of urbanization impacts rather than locality effects without collection of new human samples.

– Sharing human-like taxa supports transmission driving the dynamics (since non-urban animals miss opportunities to get microbes from any humans) while sharing only urban taxa supports ecology as well as transmission (since urban animals have more chance for transmission but also more chance of shared environmental pressures missing in rural animals). More discussion of the fact urban animals (compared to either more or most rural) are more similar to all humans is warranted. Also, a supplementary figure should be included with all human groups included in the ordination plots.

– Analyses on the differentially abundant taxa should be included which note whether they change in the other wildlife species (e.g. how does R. gnavus differ amongst anoles). This would be particularly helpful for trying to parse why the anoles and coyotes both respond to urbanization but in perpendicular direction in ordination space (distinguishing between the microbes were already elevated in rural populations so wouldn’t increase in urban or that they were missing in both).

– Text like “this convergence was evident despite the fact that the human and wildlife populations that we examined resided in different cities throughout North America, indicating parallel effects of urbanization on humans and wildlife independent of geographic location.” Needs to be adjusted to reflect the fact that the impacts of urbanization in wildlife are in fa”t not parallel (either in terms of total composition or specific taxa). Relatedly, it’s unclear in the intro if “Urbanization can alter the composition of the gut microbiota in diverse species of vertebrate wildlife (13,14), leading to consistent differences between populations in urban and rural settings” means that there are consistent differences across species or just across populations within a species.

In addition to those aforementioned weaknesses, I would like to call the authors attention to other areas which could be improved/clarified.

–The coyote urbanization dissimilarity signal seems like it could be largely driven by the one outlier which appears to the left of the humans on the ordinations. Do the findings for coyotes hold without this outlier? Relatedly, the sample set for rural coyotes is much larger than for urban ones; if only a subset of rural coyotes are included do the findings still hold?

– It is not clear how many enriched ASVs in either host species are not found in humans but it seems likely to be more than 3 or 1 (in the case of coyotes). This potentially conflicting evidence should be included alongside the positive results reported.

– What is the justification for analyzing at the ASV level? Since the human samples are not from Edmonton or Puerto Rico it seems highly unlikely there is actually strain or even ASV transmission whereas higher taxonomic level data would be more mechanistically realistic and could more strongly support effects of ecological parallels.

– What is the justification for not including phylogenetically informed dissimilarity metrics? While I am hesitant typically to just ask for more distance metrics, the authors note themselves that phylogenetic analysis would be necessary to parse sharing patterns. If the authors found weaker/absent effects in phylogenetic data how would that impact the interpretation?

– More detail should be provided on the gradient represented in the anole data and in particular how Quemado 1 and 2 should be interpreted. The data collection design for the anoles is clearly stronger than for coyotes but the nuance of the trends is not discussed proportionally.

– Why not include data from all published studies describing urbanization impacts on animals rather than just the coyotes?

Editorial changes that would improve the manuscript.

– The abstract says urban animals “acquire gut microbiota constituents found in humans” but the taxa identified in Figure 2 are also present in the non-urban populations, just at lower abundance so they clearly have not been uniquely acquired by urban populations.

– Much more detail on methods is needed. For instance, the CLR transform is not described anywhere so it’s unclear where it is being applied and to what end. How the publicly available data were accessed should be noted. IACUC information for anole collection is not included currently.

– Ideally all statistical results would be included in the text and not just figure legends. PERMANOVA r2 values should be listed somewhere.

– 2 C, D, and E and supplementary figures should all have fully abelled x axes.

– The stacked barcharts are impossible to read because colors are reused in the legend. Reduce the number of phyla plotted and bin the rest as other. Maybe have different legends for the anoles and the coyotes?

– Violin plots may be better than the box plots for dissimilarity comparisons.

– New sequencing data should be made available in a more standard repository than dryad.

*Reviewer #3 (Recommendations for the authors):*

Overall I very much enjoyed reading this paper. There were just a few places that I think that the results could be explained in a bit more context, especially regarding the relative importance of these results. Based on the title and abstract, it would be unclear whether there is complete homogenization of microbiomes among species living in cities, or whether these effects are slight. I still find these results extremely interesting, but I believe more discussion will be important so as to offer a balanced interpretation of the data.

Lines 72-73 – I am not sure of the length restrictions here, but I would have appreciated some of these supplemental results to be included in the main text. Also, can you do an analysis with effects of Species and Habitat (urban, rural) as a means of comparing which of these effects is strongest? I think this is clear from the NMDS, but I think it would be good to explicitly state the relative importance of these variables.

Line 81-82 – Should the term be pairwise distances rather than “diversities”?

Lines 86-111: Here, I think some more interpretation into the magnitude of these effects. What is the % of human associated ASVs in the urban animals? Do they make up 2% of the community? 20%? These data could be presented more explicitly to the readers. Could you try running random forest models? Would those be able to correctly assign an urban coyote sample as belonging to that group? Any other type of analysis or interpretation to better convey the “effect size” of these results are would be beneficial to the paper.

Lines 139-149 – Within the discussion of the paper I would like some acknowledgment of the limitat“ons of the ”ata. The paper opens discussing how humanization of the animal gut microbiota may cause “potentially adverse consequences for host phenotypes and fitness.” However, this idea isn’t revisited towards the end. This could be speculated on (with specific language to signal speculation), but also in a way to acknowledge that these are some of the next open research questions.

[Editors’ note: further revisions were suggested prior to acceptance, as described below.]

Thank you for resubmitting your work entitled “Humanization of wildlife gut microbiota in urban environments” for further consideration by *eLife*. Your revised article has been evaluated by George Perry (Senior Editor) and a Reviewing Editor.

The manuscript has been improved but there are some remaining issues that need to be addressed, as outlined below:

Essential revisions:

1) Reviewer 2 remains concerned about potential overstatements. Textual edits should suffice to provide context and relevant caveats.

2) Additional statistical analyses, detailed below.

*Reviewer #2 (Recommendations for the authors):*

I appreciate the additional analyses that the authors included, specifically adding further human data and another set of animal samples. However, I remain concerned that they are overstating the consistency and importance of their finding of “humanization”. There are certainly significant effects they find, but whether those add up to a general effect in urban animals is not clear to this reader. I would be hesitant to support publication without further revisions.

– The authors frequently and prominently claim to demonstrate convergence in the microbiome of urban animals, but they do not present statistics which actually robustly support this.

– They show different urban animals are more similar to urban humans but don’t actually report whether urban animals are more similar to one another (necessary for convergence). All being more similar to humans doesn’t mean they are all similar in the same way.

– Looking at the ordination plots, there doesn’t seem to be much evidence of convergence. The small shifts visible in Figure 1 FS 4 are not consistent between species such that urban animals are different from rural ones (and more similar to humans) along any of the axes. Whatever effect may be there is certainly minimal relative to the differences between species and the differences between human populations. The effect in Figure 1 D/E may be slightly stronger, but again nothing to write home about.

– The fact that the ASVs highlighted as differing in abundance (e.g. Figure 2) do not consistently differ in multiple wildlife populations or even multiple human populations further accentuates the limited scope of the urbanization effects. (It also doesn’t help that there are so few ASVs that differ and that the effect sizes for these differences are pretty small.) It would be more convincing if the ancom analyses were run with all animals together and found statistically significant results. (To a lesser extent, this problem also appear in the α diversity analyses where the human effects really only show up in Yatsunenko data but are presented as being generally true).

– Similarly, the random forest models on each species don’t tell us much, whereas a model which could accurately predict across species would be pretty convincing.

– The authors attempted in their revision to better dissect what may cause the patterns they observe, but still err too much on the side of human transmission. In particular, they seem to believe the animals and humans being from different sites is enough to rule out animal to human transmission driving the patterns (see lines 150-156) but it’s unclear to me why that doesn’t also apply to human to wildlife transmission. It seems possible that if the urban environment had consistent effects on animal microbiomes in different locales, they could converge and then spread microbes to humans. The evidence here doesn’t support that per se, but it also doesn’t clearly support human to animal transmission.

Presentation of methods/results remains suboptimal.

– While PERMANOVA values were added to the text, no other statistical results are cited in line which makes it hard to determine the strength of their claims (especially when they use terms like “tended” (line 117)).

– There are insufficient methodological details in the text and the GitHub page has not been updated since the revision making it hard to tell how exactly tests were carried out. Most importantly, no details are provided on the random forest models in the methods text and the structure of the adonis models is not specified.

– Finally, in line with the authors overstatement of results, the introduction reads as biased towards credulity over generalizable impacts of humans on animal microbiomes.

– It is true that captivity frequently alters the microbiome but it is just as important to note that it rarely does so in consistent or convergent ways (see Diaz and Reese 2021 Animal Microbiome for a review and Alberdi et al., 2021 Sci Reports for a meta-analysis). There are only a handful of studies which show the gradient effects so prominently highlighted here.

– Moreover, the functional implications of these changes are unknown in the vast majority of cases. Neither ref 7 or 14 include evidence that altered microbiomes cause disease, they just suggest it. There are a few papers which use FMTs to show improvement in captive animal condition but mostly this idea remains untested (see again Diaz and Reese 2021 for a discussion of the literature).

– It seems odd to not also discuss the effects of domestication on microbiomes since that is a system where much more is known about transmission between animals and humans.

*Reviewer #3 (Recommendations for the authors):*

I am happy with the revisions conducted here. The inclusion of additional datasets strengthened the conclusions, including in some other wild vertebrate clades. The authors have done a nice job including the other recommendations as well. Very interesting study!

---

## [Author Response]

Essential revisions:1) Expand the human datasets analyzed. The current data doesn't seem that well suited to look at urbanization vs. other confounding factors. If possible, it would be great to include other wildlife urban/rural comparisons and other human datasets which more explicitly get at urbanization. Potential comparator datasets include:https://link.springer.com/article/10.1007/s00248-020-01569-8https://www.sciencedirect.com/science/article/abs/pii/S0048969717323781https://onlinelibrary.wiley.com/doi/full/10.1111/mec.15240https://pubmed.ncbi.nlm.nih.gov/30439937/2) Address the various other points itemized below, including addressing the limitations and caveats of these data.

Thank you for these excellent suggestions. We have added several new datasets to the manuscript, all of which validated the conclusions in the initial submission. These datasets include white-crowned sparrows collected from urban and rural settings in and around San Francisco, California, USA; an urban-to-rural transect of humans living in Cameroon; an urban-to-rural transect of humans living Tibet; and additional data generated by us from other *Anolis* species residing in the rural sampling site of Maricao. Note that although these new anole data were derived from non-*cristatellus* species, their phylogenetic relatedness to humans is equal to that of *cristatellus*, providing another point of comparison to test the hypothesis that urban settings lead to increased gut-microbiota similarity with humans. The additional human and wildlife datasets were chosen because they (1) used the same forward V4 16S primer as the datasets from the initial submission and (2) had sufficient sample sizes from rural and urban populations (>5 samples per population after quality/read-depth filtering).

The overall approach and conclusions of the study remained unchanged in this revision compared to the initial submission. However, the new data increased the number of comparisons capable of testing our hypothesis by several fold. All results supported the conclusion that human and wildlife gut microbiota have converged in cities relative to rural populations. These additions have substantially strengthened the results in manuscript (e.g., Figure 1) compared to the initial submission. Another new result of particular interest is that the *Bacteroides* ASV most overrepresented in urban anoles relative to rural anoles shows the parallel shift in abundance in all three human datasets, and significant parallel shifts in two of the human datasets (in fact, this ASV was remarkably the top Urban-associated hit of all ASVs in two of the human datasets, which were generated from samples from different continents).

We have rewritten the manuscript to incorporate these additions, and we have addressed all of the comments from the Reviewers.

Reviewer #1 (Recommendations for the authors):As mentioned in the discussion, the addition of metagenomics would greatly strengthen the conclusions that can be made. Alternatively, the authors could use qPCR, FISH, selective culture, or other methods to strengthen the interpretation that the same bacteria are shared between host species.

We agree that metagenomics or other analyses at the sub-ASV level would provide higher resolution to detect strain sharing in urban environments. We have revised our discussion of potential future directions to more clearly address this point. However, as we note above, the distinctiveness of the 16S rDNA profiles of rural anoles/coyotes/sparrows and humans enabled the use of 16S rDNA sequencing to assess the hypothesis that microbiota of wildlife and humans have converged in cities. Our results cannot be explained by limitations of 16S rDNA sequencing (although in the event that we observed null results, these could have been attributed to lack of resolution afforded by 16S rDNA sequencing). Overall, our results provide a lower bound for the number of independent instances of lineages shared by urban wildlife and humans but not by rural wildlife, as each ASV showing the signal of urbanization may in fact contain multiple sub-ASV lineages.

It is also important to rule out any technical explanations for overlap in ASV content between species, including barcode hopping, cross-contamination, or other issues. More discussion should be included as to how these samples were sequenced and what control samples were included to rule out these artifacts.

We have added sentences to the main text to better clarify the studies from which the data were derived. The independent datasets analyzed could not have been cross contaminated by barcode hopping, as each dataset was prepared and sequenced separately. With the inclusion of multiple additional datasets, each of which supports the conclusion of wildlife-human microbiota convergence in cities, there are no known technical issues that can explain the results presented.

There seems to be a conceptual issue in the degree to which one should be surprised to find overlapping bacterial species between host species. As Drs. Moeller, Ley, and others have shown, the phylogenetic signal requires sophisticated analyses and is only true for subsets of the gut microbiota. It would help to assess the range of ASV overlap between humans and other species to see if the observed hit rate in this study would be expected by chance.

The design of our study included internal controls for each wildlife urban population (i.e., the rural wildlife conspecific populations) to explicitly test the idea posed by the reviewer. We found that urban wildlife and humans harbored significantly more similar microbiota than did rural wildlife and humans. These results cannot be explained by chance overlap, or a baseline level of overlap between vertebrate microbiota (such as that observed between rural wildlife and humans).

I also didn't follow the logic that transfer events would result in differential abundance as a function of urbanization. Isn't it possible that bacteria could be consistently transferred from humans to coyotes in both urban and rural settings? What if a transfer event only happens in urban settings but is rare, resulting in a non-significant p-value? It's important to provide a clearer rationale for why each analysis was done and whether it answers the hypothesis.

We have clarified the rationale for the ANCOM analyses in the main text. The central ideas of these analyses were to test (1) whether certain ASVs were consistently differentially abundant between urban and rural individuals within individual host species/human datasets and (2) whether the ASVs that were differentially abundant between urban and rural individuals in at least one wildlife species and one human dataset showed the same direction of difference (over- or under-represented in urban environments) in both wildlife and humans. These results provided significant support for parallel shifts in the relative abundance of ASVs in urban humans and urban wildlife. These patterns could result from altered transmission within urban environments or by urban-specific selective agents. We now better articulate these non-mutually exclusive explanations in the discussion.

It is possible that “bacteria could be consistently transferred from humans to coyotes in both urban and rural settings”. However, that urban wildlife and humans harbored more similar sets of ASVs than did rural wildlife and humans supports the conclusion that sharing of microbiota between humans and wildlife is elevated in urban settings relative to rural settings (which could result from transmission or selection).

The question “What if a transfer event only happens in urban settings but is rare, resulting in a non-significant p-value?”: raises an excellent point. We now clarify in the supplemental materials that false negatives are expected in these analyses. However, we note that false negatives cannot explain the primary results that ASVs display parallel shifts between urban and rural populations across host species.

Reviewer #2 (Recommendations for the authors):Here I note some particular ways to address shortcomings noted in the Public Review.– The human data used here were originally collected to analyze industrialization effects rather than urbanization effects and span populations that vary in many dimensions. While obviously related urbanization and industrialization transitions are distinct so it is impossible to say that urbanization is really what underlies the effects here. No good datasets exist comparing urban and rural populations in North America (which would be ideal because that is where all the wildlife data is from) but there are within country analyses for Cameroon (Lokmer et al., Sci Rep 2020), Nigeria (Ayeni et al., Cell Rep 2018), and India (Das et al., Sci Rep 2018). Presumably if the urbanization signal is thought to be generalizable it should still show up in studies from outside North America so why not use those? Less ideal but perhaps relevant are the gradient of industrialization presented in Jha et al., (2018 PLoS Bio) and the comparison of Indigenous Canadians in Montreal and Inuit villages (Girard et al., mSphere 2017) at least control for geography when investigating lifestyle transitions. Perhaps some combination of these datasets would allow for a more robust test of urbanization impacts rather than locality effects without collection of new human samples.

Again, thank you for this excellent suggestion. Unfortunately, several of the datasets suggested by the Reviewer used different 16S rDNA primers, precluding direct comparisons of these data with the wildlife data presented. However, some of the datasets suggested and others in the literature did use the V4 forward primer, enabling us to include them in our analyses. Reperforming all tests in the initial submission with these new data (and another wildlife dataset from rural and urban sparrows) greatly increased support for our conclusions of convergence of wildlife and human microbiota in urban settings. These are particularly insightful, as suggested by the reviewer, because they revealed a new finding: elevated microbiota overlap between urban wildlife and urban humans relative to urban wildlife and ‘semi-urban’ (i.e., suburban) humans. This new result lends further support to the conclusion that convergence of human and wildlife gut microbiota is particularly elevated in urban settings.

– Sharing human-like taxa supports transmission driving the dynamics (since non-urban animals miss opportunities to get microbes from any humans) while sharing only urban taxa supports ecology as well as transmission (since urban animals have more chance for transmission but also more chance of shared environmental pressures missing in rural animals). More discussion of the fact urban animals (compared to either more or most rural) are more similar to all humans is warranted. Also, a supplementary figure should be included with all human groups included in the ordination plots.

We have added the suggested supplemental figure showing all the data in ordination plots. Moreover, we have expanded our discussion to better articulate the alternative processes that may be underlying the observed convergence of urban wildlife and human gut microbiota (relative to rural wildlife). This discussion is also described in our responses to Reviewer 1 above.

– Analyses on the differentially abundant taxa should be included which note whether they change in the other wildlife species (e.g. how does R. gnavus differ amongst anoles). This would be particularly helpful for trying to parse why the anoles and coyotes both respond to urbanization but in perpendicular direction in ordination space (distinguishing between the microbes were already elevated in rural populations so wouldn't increase in urban or that they were missing in both).

We now include in the Supplementary Files lists of all the significant ASVs identified in these analyses to enable identification of ASVs displaying shared or not shared responses to urbanization in different wildlife species. None of the ASVs were significant in multiple wildlife species, although we note that these analyses were likely underpowered (as noted by Reviewer 1) (however, also note that the positive results from these analyses cannot be explained by lack of power – thus, our results provide a lower bound on the number of ASVs displaying parallel shifts among wildlife and human populations).

– Text like "this convergence was evident despite the fact that the human and wildlife populations that we examined resided in different cities throughout North America, indicating parallel effects of urbanization on humans and wildlife independent of geographic location." needs to be adjusted to reflect the fact that the impacts of urbanization in wildlife are in fact not parallel (either in terms of total composition or specific taxa). Relatedly, it's unclear in the intro if "Urbanization can alter the composition of the gut microbiota in diverse species of vertebrate wildlife (13,14), leading to consistent differences between populations in urban and rural settings" means that there are consistent differences across species or just across populations within a species.

Thank you for noting these unclear and potentially misleading statements. We have edited these sentences to clarify that our results indicate microbiota convergence between wildlife and humans in urban environments, but that the specific ASVs underlying the convergence differed among wildlife species.

In addition to those aforementioned weaknesses, I would like to call the authors attention to other areas which could be improved/clarified.–The coyote urbanization dissimilarity signal seems like it could be largely driven by the one outlier which appears to the left of the humans on the ordinations. Do the findings for coyotes hold without this outlier? Relatedly, the sample set for rural coyotes is much larger than for urban ones; if only a subset of rural coyotes are included do the findings still hold?

We have repeated all test of microbiota convergence without the coyote sample identified. Results indicated that the convergence remained significant in every case (p<0.01) even without this sample. We now report these results in the text. Regarding the uneven sampling effort for rural and urban coyotes, this is not expected to yield false positives in either direction (i.e., increased overlap or decreased overlap). However, to address this point, we repeated the tests of microbiota convergence with five random subsamples of 50% of the rural coyote samples. Each of these subsets of the data also supported the original conclusions in every case (p<0.01).

– It is not clear how many enriched ASVs in either host species are not found in humans but it seems likely to be more than 3 or 1 (in the case of coyotes). This potentially conflicting evidence should be included alongside the positive results reported.

We now include lists of all ASVs overrepresented between urban and rural populations for each host species in the Supplemental Materials. There were ASVs overrepresented in urban wildlife relative to rural wildlife that were not found in humans, but we note that this result does not contradict the primary result (i.e., of the ASVs that were shared by wildlife and humans and differentially abundant in urban vs rural populations of each species, all of these ASVs showed the same direction of abundance shift in both humans and wildlife).

– What is the justification for analyzing at the ASV level? Since the human samples are not from Edmonton or Puerto Rico it seems highly unlikely there is actually strain or even ASV transmission whereas higher taxonomic level data would be more mechanistically realistic and could more strongly support effects of ecological parallels.

Thank you for noting the lack of justification for this choice in the initial submission. As noted in the Reviewer’s previous comment and by Reviewer 1, sharing of microbial lineages between populations is best assessed by fine-scale analyses of microbiota content (i.e, below the 97% OTU or genus level). Therefore, we focused all analyses on the finest resolution afforded by our data (i.e., 100% 16S rDNA ASVs). Estimates for the rate of sequence evolution of 16S rDNA sequences in bacteria indicate a rate of ~1% per 50 million years (eg as estimated by insect endosymbionts and enterics in mammals). Given this slow rate of 16S rDNA evolution, identical ASVs are found across all human populations included in this study despite the geographic diversity. This enabled us to test for evidence for human derived microbes in urban wildlife populations even though the humans and wildlife were not sampled in the same cities. Analyses at higher taxonomic scales suffer from reduced power to identify signal of sharing of individual microbial lineages between urban wildlife and humans, because many of the genera and higher taxonomic ranks are shared across all populations (including rural populations).

– What is the justification for not including phylogenetically informed dissimilarity metrics? While I am hesitant typically to just ask for more distance metrics, the authors note themselves that phylogenetic analysis would be necessary to parse sharing patterns. If the authors found weaker/absent effects in phylogenetic data how would that impact the interpretation?

We now clarify this choice in the text. Phylogenetic dissimilarity measures would not be sensitive to acquisition by urban wildlife of human-derived lineages within clades of microbes already present in rural wildlife. For example, the acquisition of human-specific *Ruminococcus* ASVs would have minimal effects on Unifrac dissimilarity between coyotes and humans, because coyotes already harbor other *Ruminococcus* lineages (i.e., acquiring an additional *Ruminococcus* lineage would only add a small amount of new branch length to the coyote-gut microbiota phylogeny). Given the taxonomic similarity between human and wildlife gut microbiota at the genus level and above, non-phylogenetic dissimilarity measures are better suited to test for increased sharing of lineages (i.e., more similar community memberships) between urban wildlife and humans relative to rural wildlife.

– More detail should be provided on the gradient represented in the anole data and in particular how Quemado 1 and 2 should be interpreted. The data collection design for the anoles is clearly stronger than for coyotes but the nuance of the trends is not discussed proportionally.

Thank you for this suggestion. We have included additional discussion of the anole transect and how the results observed suggest that distance to urban environments is negatively associated with the degree of microbiota overlap with humans.

– Why not include data from all published studies describing urbanization impacts on animals rather than just the coyotes?

Unfortunately, many of the available datasets used different 16S rDNA primers, precluding their inclusion in this study. However, we were able to include a recently published dataset from urban and rural sparrows. Interestingly, this dataset also indicated that gut microbiota from urban sparrow populations were more similar to human gut microbiota than were the gut microbiota from rural sparrow populations. This new results adds another line of evidence in support of the primary conclusions of the manuscript, and suggests that the patterns reported may be generalizable (although we do not speculate on this point in the text).

Editorial changes that would improve the manuscript.– The abstract says urban animals "acquire gut microbiota constituents found in humans" but the taxa identified in Figure 2 are also present in the non-urban populations, just at lower abundance so they clearly have not been uniquely acquired by urban populations.

We now include a list of all ASVs shared uniquely by urban wildlife and humans to better test/support this conclusion in the abstract. More ASVs were shared by urban wildlife and humans but not found in rural wildlife than were shared by rural wildlife and humans but not found in urban wildlife. Moreover, the microbiota convergence results between urban wildlife and humans shown in Figure 1 also reflect the elevated levels of sharing of ASVs between these hosts.

– Much more detail on methods is needed. For instance, the CLR transform is not described anywhere so it's unclear where it is being applied and to what end. How the publicly available data were accessed should be noted. IACUC information for anole collection is not included currently.

We have added additional details about the ANCOM methods and IACUC to the Supplementary Material to address these issues. We note that the CLR transform used was implemented in ANCOM using default settings.

– Ideally all statistical results would be included in the text and not just figure legends. PERMANOVA r2 values should be listed somewhere.

We now include the R^2^ and p-values of PERMANOVA as well as statistics for individual β diversity comparisons in the text.

– 2 C, D, and E and supplementary figures should all have fully labeled x axes.

We have added these labels.

– The stacked barcharts are impossible to read because colors are reused in the legend. Reduce the number of phyla plotted and bin the rest as other. Maybe have different legends for the anoles and the coyotes?

We have edited the color palette. In addition, we provide the raw ASV table and taxonomy assignments in the supplement to enable easier investigation of the taxonomic assignments for interested readers.

– Violin plots may be better than the box plots for dissimilarity comparisons.

We remade Figure 1 with violin plots following the Reviewer’s suggestion. However, this figure was visually difficult to interpret given all the additional comparisons included in this revision. We now include all ASV counts and dissimilarity statistics in the revision to provide alternatives to the visualization in Figure 1.

– New sequencing data should be made available in a more standard repository than dryad.

We have uploaded the data to ENA under accession PRJEB51262.

Reviewer #3 (Recommendations for the authors):Overall I very much enjoyed reading this paper. There were just a few places that I think that the results could be explained in a bit more context, especially regarding the relative importance of these results. Based on the title and abstract, it would be unclear whether there is complete homogenization of microbiomes among species living in cities, or whether these effects are slight. I still find these results extremely interesting, but I believe more discussion will be important so as to offer a balanced interpretation of the data.

Thank you for noting this important issue, as the main text in the initial submission lacked a clear statement about the effects of host-species identity on microbiota composition. We have added statements to the text clarifying the magnitude of the effects observed, including that host species identity was a major determinant of microbiota composition in both urban and rural populations.

Lines 72-73 – I am not sure of the length restrictions here, but I would have appreciated some of these supplemental results to be included in the main text. Also, can you do an analysis with effects of Species and Habitat (urban, rural) as a means of comparing which of these effects is strongest? I think this is clear from the NMDS, but I think it would be good to explicitly state the relative importance of these variables.

We have added two supplemental tables that display all pairwise β diversity comparisons between pairs of sample groups. In addition, we now include more details about the results of the PERMANOVA analyses. These suggested that the effect of Urban/Rural status on β diversity was less than that of host species identity, but comparable to effects of location/population within host species. We have added statements to the text indicating that host species identity remained the predominant explanatory variable for microbiota composition for most samples regardless of environment.

Line 81-82 – Should the term be pairwise distances rather than "diversities"?

We have replaced ‘diversities’ with ‘pairwise dissimilarities’.

Lines 86-111: Here, I think some more interpretation into the magnitude of these effects. What is the % of human associated ASVs in the urban animals? Do they make up 2% of the community? 20%? These data could be presented more explicitly to the readers. Could you try running random forest models? Would those be able to correctly assign an urban coyote sample as belonging to that group? Any other type of analysis or interpretation to better convey the "effect size" of these results are would be beneficial to the paper.

Although our results provide strong evidence for increased microbiota sharing by wildlife and humans in urban settings, the effects of urbanization were weaker than those of host species identity. We now make this finding clear in the text to avoid possible misinterpretation of our results. We also present in the Supplementary Files a list of ASVs shared by urban wildlife and humans but not be rural wildlife to provide more details about the total number of ASVs driving the patterns observed.

In addition, we have added a supplemental figure showing the accuracy of random forest classifiers trained on the anole, coyote, and sparrow data, as suggested (these used 50% of the data to train and 50% to test). These classifiers were able to accurately classify wildlife into urban or rural populations beyond what would be expected by chance.

Lines 139-149 – Within the discussion of the paper I would like some acknowledgment of the limitations of the data. The paper opens discussing how humanization of the animal gut microbiota may cause "potentially adverse consequences for host phenotypes and fitness." However, this idea isn't revisited towards the end. This could be speculated on (with specific language to signal speculation), but also in a way to acknowledge that these are some of the next open research questions.

We have added in the main text a statement about the need for future work to address the phenotype and fitness consequences of spillover of human microbiota in wildlife populations. In addition, we have edited the introduction to better explain the rationale for why interspecific transmission of microbiota may adversely affect host fitness. However, due to space limitations (unfortunately, the main text is already slightly over the word limit) and lack of relevant data, we have limited the speculation in the text and focus instead on suggestions for future work.

[Editors’ note: further revisions were suggested prior to acceptance, as described below.]

Essential revisions:1) Reviewer 2 remains concerned about potential overstatements. Textual edits should suffice to provide context and relevant caveats.

Thank you for noting these issues. We have corrected and clarified several statements throughout the text to address points raised by Reviewer #2. In addition, we have added the suggested analysis to test for convergence of the gut microbiota of urban populations of different wildlife species (although we note that this analysis does not directly address the focus of this Short Report regarding whether urban wildlife gut microbiota have converged with human gut microbiota relative to rural wildlife gut microbiota). The new analysis is included in this revision as Figure 1—figure supplement 6. Results indicated that gut microbiota of different wildlife species were more similar in urban environments than between urban and rural environments in two out of the three comparisons of pairs of wildlife species. This addition further supports that urban environments alter wildlife gut microbiota, but the primary conclusion of the manuscript pertaining to microbiota convergence with humans is not affected.

The primary finding of this study that wildlife microbiota converged with human microbiota in cities relative to rural wildlife microbiota was a generalizable finding well-supported across all combinations of wildlife and human microbiota examined. This pattern has not been described previously to our knowledge and is suggestive of transmission of bacteria between humans and wildlife (most likely from humans into wildlife) and/or parallel selection on human and wildlife microbiota in cities. Importantly, this primary result highlights the need for future studies of urban wildlife microbiota to consider the possibility of increased microbiota sharing with humans (as this appears to be an at least somewhat widespread pattern--i.,e., it was observed across all datasets examined).

2) Additional statistical analyses, detailed below.

We have included several additional statistical analyses as suggested by Reviewer #2. In particular, the random-forest model and ANCOM analysis provided additional support that populations could be differentiated based on urban vs rural status. However, as above, we note that the primary focus of this Short Report is the convergence of wildlife microbiota with human microbiota in urban environments relative to rural wildlife microbiota. The new analyses, although interesting, do not necessarily address the primary conclusions of the Short Report, so we therefore include them in the supplement with brief mention in the main text.

In addition, we thank Reviewer #2 for the suggestion to include statistics and p-values in the text of the manuscript (rather than in figures and tables alone), which has been revised accordingly. In cases where many tests were performed such that listing all statistics in the text was not practical, we include these in supplementary data files, tables, figures, and figure legends.

Reviewer #2 (Recommendations for the authors):I appreciate the additional analyses that the authors included, specifically adding further human data and another set of animal samples. However, I remain concerned that they are overstating the consistency and importance of their finding of "humanization". There are certainly significant effects they find, but whether those add up to a general effect in urban animals is not clear to this reader. I would be hesitant to support publication without further revisions.

Thank you for your time in again carefully reviewing our revised manuscript and for the additional suggestions for improvement. In this revision, we have clarified presentation of our results, particularly the finding that wildlife microbiota converged with human microbiota in cities relative to rural wildlife microbiota. We have also added several new analyses, including (1) tests of whether wildlife microbiota of different species converged with one another in urban environments, (2) the suggested random-forest analysis, and (3) the suggested ANCOM analysis. Results are described below. These additions have improved the study. The primary conclusion of this manuscript that urban wildlife microbiota were significantly more similar to human microbiota than were rural wildlife microbiota remains well-supported. This was a general finding in that it was apparent in all of the wildlife species examined in the context of multiple human datasets spanning rural to urban transitions.

– The authors frequently and prominently claim to demonstrate convergence in the microbiome of urban animals, but they do not present statistics which actually robustly support this.

We did not test in the previous submission whether urban wildlife microbiota converged with one another relative to rural wildlife microbiota. Rather, our results showed that urban wildlife gut microbiota converged with human microbiota relative to rural conspecific wildlife (i.e., rural wildlife and human microbiota were more similar than were rural wildlife and human microbiota). The statistics presented in Figure 1, the Main Text, and the Supplementary Materials strongly support this conclusion. We have added clarification to the Main Text that we use the term ‘convergence’ to refer to cases in which urban wildlife microbiota and human microbiota were more similar than were rural wildlife microbiota and human microbiota.

In addition, we have added a new analysis that addresses the evidence for ‘convergence in the microbiome of urban animals’ (Figure 1—figure supplement 6). In this analysis, we tested whether microbiota similarity between urban populations of pairs of wildlife species was higher than that between urban and rural populations of the wildlife species. Significantly increased microbiota similarity between urban populations was observed in two out of three comparisons of pairs of wildlife species (anoles vs. sparrows and coyotes vs. anoles). These results support that some of the microbiota shifts in urban environments were shared among wildlife species but do not directly pertain to whether urban wildlife microbiota converged with human microbiota relative to rural wildlife microbiota.

– They show different urban animals are more similar to urban humans but don't actually report whether urban animals are more similar to one another (necessary for convergence). All being more similar to humans doesn't mean they are all similar in the same way.

Given the multivariate nature of microbiome datasets, it is possible for urban wildlife and human gut microbiota to display convergence in cities relative to rural wildlife without the microbiota of urban wildlife populations displaying evidence of convergence with one another relative to rural wildlife. We have rewritten sections of the main text to better clarify precisely what can be concluded from our results.

However, as noted above, we conducted an analysis to directly assess whether urban animal microbiota were more similar to one another (Figure 1—figure supplement 6). Results provided some evidence for convergence among microbiota of urban wildlife species, but this finding does not alter the primary conclusion of the manuscript regarding convergence between wildlife and human microbiota in cities.

– Looking at the ordination plots, there doesn't seem to be much evidence of convergence. The small shifts visible in Figure 1 FS 4 are not consistent between species such that urban animals are different from rural ones (and more similar to humans) along any of the axes. Whatever effect may be there is certainly minimal relative to the differences between species and the differences between human populations. The effect in Figure 1 D/E may be slightly stronger, but again nothing to write home about.

Ordination plots, although useful tools for visualization, are not able to provide statistical support for the conclusions of the manuscript. For example, the apparently large differences among the human populations in the ordination plots, as noted by the Reviewer, reflects that humans represent a greater % of the total number of samples in the dataset than wildlife species rather than elevated β diversity between human populations. We caution against over interpreting these PCoA plots, which are merely visualization tools rather than definitive displays of the underlying β diversity measures. That the convergence of urban wildlife and human microbiota was evident in these plots (eg Figure 1D/E) highlights the observed effects, but does not provide quantitative statistical evidence for convergence. The latter comes from tests for differences in β diversity between pairs of groups (i.e., using the β diversity matrix directly rather than the ordination axes) as shown in Figure 1 panels A–C. These panels demonstrate that urban wildlife microbiota were significantly more similar to human microbiota in urban environments relative to rural wildlife microbiota.

– The fact that the ASVs highlighted as differing in abundance (e.g. Figure 2) do not consistently differ in multiple wildlife populations or even multiple human populations further accentuates the limited scope of the urbanization effects. (It also doesn't help that there are so few ASVs that differ and that the effect sizes for these differences are pretty small.) It would be more convincing if the ancom analyses were run with all animals together and found statistically significant results. (To a lesser extent, this problem also appear in the α diversity analyses where the human effects really only show up in Yatsunenko data but are presented as being generally true).

Thank you for these comments. As we note above, the conclusions of the manuscript do not necessitate that the same ASVs underlie the convergence with human microbiota in cities in every wildlife species examined. We find that several prominent ASVs display parallel shifts in abundances between urban versus rural wildlife and urban versus rural humans.

The suggested analyses, in which ANCOM is rerun with all wildlife species together, addresses a different question from those originally posed in the manuscript. Namely, this suggested analysis is well-suited to test whether any ASVs show consistent shifts in relative abundances across all urban vs rural wildlife comparisons. We have conducted the suggested analysis—i.e., ANCOM to test for differential abundance between all rural and urban wildlife populations using host species as a covariate—and now report the results in Figure 2—figure supplement 4 and Supplementary File 7. Results show that several ASVs remain significant even when considering all animal species simultaneously. Interestingly, the *Bacteroides* ASV identified as overrepresented in urban anoles and urban human populations also displayed significant overrepresentation in this analysis including all wildlife species.

In addition, we have revised the sentence pertaining to the α diversity results to better reflect that the differences in α diversity between rural and urban human populations did not reach statistical significance in all datasets (The most significant difference was observed in the Yatsunenko dataset, which contained the largest number of samples, but the direction of difference was shared by all other human datasets).

– Similarly, the random forest models on each species don't tell us much, whereas a model which could accurately predict across species would be pretty convincing.

The random forest models trained on individual species, as suggested by Reviewer #3, are informative in that these analyses indicate that conspecific rural and urban populations of wildlife can be differentiated from one another through this sample classifier approach. In addition, we have conducted the suggested analysis including all animal species and humans, using ‘Rural’ versus ‘Urban’ as the discriminatory variable. This analysis showed that the classifier was able to differentiate ‘Rural’ and ‘Urban’ samples even when all animal species were included in a single analysis. In fact, the accuracy of the model trained on all host taxa simultaneously performed better than any model trained on an individual host taxon. These results lend further support towards the conclusions that rural and urban populations display different microbiota profiles.

– The authors attempted in their revision to better dissect what may cause the patterns they observe, but still err too much on the side of human transmission. In particular, they seem to believe the animals and humans being from different sites is enough to rule out animal to human transmission driving the patterns (see lines 150-156) but it's unclear to me why that doesn't also apply to human to wildlife transmission. It seems possible that if the urban environment had consistent effects on animal microbiomes in different locales, they could converge and then spread microbes to humans. The evidence here doesn't support that per se, but it also doesn't clearly support human to animal transmission.

The reasoning underlying the statements about the likelihoods of human to wildlife transmission vs. wildlife to human transmission was not only based on the fact that the animals and humans were from different sites (lines 150–156), but also that the wildlife species examined here do not exist where several of the human populations reside (i.e., the wildlife ranges do not include the locations where some of the human populations were sampled, e.g., Tibet). In contrast, humans are present at all of the wildlife sampling locations. We reasoned that human to wildlife transmission is a more likely explanation for the patterns observed than wildlife to human transmission (e.g., it is not clear how anoles would transmit microbes to humans living in Tibet, where none of these wildlife species are found). The potential alternative stated in the Main Text is at least less parsimonious than the explanation of human-to-wildlife transmission. We have clarified this statement in the text to avoid potentially misleading readers. In addition, we have edited the text since the initial submission to better reflect that we have not demonstrated any specific transmission events, and that alternative explanations for our results (e.g., parallel dietary shifts) have not been falsified.

Presentation of methods/results remains suboptimal.– While PERMANOVA values were added to the text, no other statistical results are cited in line which makes it hard to determine the strength of their claims (especially when they use terms like "tended" (line 117)).

Thank you for noting these omissions, particularly in the context of the vague wording on line 117. We have added statistics and p-values with multiple sentences throughout the text. For many tests the large number of comparisons preclude listing all statistics in the text, so we have instead included these in supplemental tables and/or figures.

– There are insufficient methodological details in the text and the GitHub page has not been updated since the revision making it hard to tell how exactly tests were carried out. Most importantly, no details are provided on the random forest models in the methods text and the structure of the adonis models is not specified.

Thank you for noting these omissions. We have updated the GitHub page and added details about methods in the text. In particular, we now include the structure of the adonis2 models and additional details about the random forest models in the Materials and methods.

– Finally, in line with the authors overstatement of results, the introduction reads as biased towards credulity over generalizable impacts of humans on animal microbiomes.

We have edited the manuscript to clarify our conclusions and avoid overstating results or misleading readers. We have clarified in this revision that our results support that urban wildlife microbiota have converged with human microbiota in cities relative to rural wildlife microbiota (as opposed to convergence of all the urban wildlife microbiota to one another). We have also edited the introduction to better address the points raised by Reviewer #2 in this round of revision.

– It is true that captivity frequently alters the microbiome but it is just as important to note that it rarely does so in consistent or convergent ways (see Diaz and Reese 2021 Animal Microbiome for a review and Alberdi et al., 2021 Sci Reports for a meta-analysis). There are only a handful of studies which show the gradient effects so prominently highlighted here.

We have made several edits in the introduction and throughout the text to clarify that urbanization (or captivity) does not necessarily lead to shifts in the same bacterial ASVs or other taxa in all wildlife species. Instead, we find that, for the most part, different sets of ASVs underly convergence of urban wildlife microbiota with human microbiota relative to rural wildlife microbiota in each of the wildlife species examined (as noted in the previous round of review; although the new ANCOM analyses show that a minority of ASVs displayed significantly differential abundances when considering all wildlife species simultaneously). These changes do not alter the primary conclusions of the manuscript.

In addition, we have included the suggested citations to Diaz and Reese 2021 and Alberdi et al., 2021 in addition to a statement in the introduction that the effects of captivity on the microbiota vary among host taxa. Interestingly, one of the more generalizable findings of Alberdi et al., 2021 was an enrichment of human-associated microbes in captive mammals relative to wild mammals (although the specific microbial taxa driving this pattern differed among captive mammal taxa).

– Moreover, the functional implications of these changes are unknown in the vast majority of cases. Neither ref 7 or 14 include evidence that altered microbiomes cause disease, they just suggest it. There are a few papers which use FMTs to show improvement in captive animal condition but mostly this idea remains untested (see again Diaz and Reese 2021 for a discussion of the literature).

We have added the suggested citations regarding FMTs and the review paper, which provide more appropriate references for our discussion of the possibility that humanization of the gut microbiota in captivity may be deleterious for hosts. We agree that the health effects of microbiota changes in captivity remain to be definitively established.

– It seems odd to not also discuss the effects of domestication on microbiomes since that is a system where much more is known about transmission between animals and humans.

We have added citation to Reese et al., 2021 *eLife* regarding the effects of domestication on the microbiota. Although the current manuscript is not focused on domestication, there are parallels between the studies worth referencing.

Reviewer #3 (Recommendations for the authors):I am happy with the revisions conducted here. The inclusion of additional datasets strengthened the conclusions, including in some other wild vertebrate clades. The authors have done a nice job including the other recommendations as well. Very interesting study!

Thank you for your time and effort in reviewing the manuscript and for the constructive feedback.